# MUC1-C Dependence for the Progression of Pancreatic Neuroendocrine Tumors Identifies a Druggable Target for the Treatment of This Rare Cancer

**DOI:** 10.3390/biomedicines12071509

**Published:** 2024-07-08

**Authors:** Hiroki Ozawa, Naoki Haratake, Ayako Nakashoji, Tatsuaki Daimon, Atrayee Bhattacharya, Keyi Wang, Keisuke Shigeta, Atsushi Fushimi, Kazumasa Fukuda, Yohei Masugi, Ryo Yamaguchi, Minoru Kitago, Hirofumi Kawakubo, Yuko Kitagawa, Donald Kufe

**Affiliations:** 1Dana-Farber Cancer Institute, Harvard Medical School, 450 Brookline Avenue, D830, Boston, MA 02215, USA; hiroki_ozawa@dfci.harvard.edu (H.O.); naoki_haratake@dfci.harvard.edu (N.H.); ayako_nakashoji@dfci.harvard.edu (A.N.); tatsuaki_daimon@dfci.harvard.edu (T.D.); atrayee_bhattacharya@dfci.harvard.edu (A.B.); keyi_wang@dfci.harvard.edu (K.W.); keisuke_shigeta@dfci.harvard.edu (K.S.); atsushi_fushimi@dfci.harvard.edu (A.F.); 2Department of Surgery, Keio University School of Medicine, Shinjuku-ku, Tokyo 160-8582, Japan; fukudak@z6.keio.jp (K.F.); gupiogupio@gmail.com (R.Y.); dragonpegasus@keio.jp (M.K.); hkawakubo@z3.keio.jp (H.K.); kitagawa@a3.keio.jp (Y.K.); 3Division of Diagnostic Pathology, Keio University School of Medicine, Shinjuku-ku, Tokyo 160-8582, Japan; yohei_masugi@dfci.harvard.edu

**Keywords:** MUC1-C, pNET, MYC, mTOR, NOTCH2, CSC

## Abstract

Patients with pancreatic neuroendocrine tumors (pNETs) have limited access to effective targeted agents and invariably succumb to progressive disease. MUC1-C is a druggable oncogenic protein linked to driving pan-cancers. There is no known involvement of MUC1-C in pNET progression. The present work was performed to determine if MUC1-C represents a potential target for advancing pNET treatment. We demonstrate that the *MUC1* gene is upregulated in primary pNETs that progress with metastatic disease. In pNET cells, MUC1-C drives E2F- and MYC-signaling pathways necessary for survival. Targeting MUC1-C genetically and pharmacologically also inhibits self-renewal capacity and tumorigenicity. Studies of primary pNET tissues further demonstrate that MUC1-C expression is associated with (i) an advanced NET grade and pathological stage, (ii) metastatic disease, and (iii) decreased disease-free survival. These findings demonstrate that MUC1-C is necessary for pNET progression and is a novel target for treating these rare cancers with anti-MUC1-C agents under clinical development.

## 1. Introduction

Neuroendocrine tumors (NETs) are a heterogeneous class of cancers that commonly originate in the pancreas [1]. The annual incidence of pancreatic NETs (pNETs) is approximately 1–5 in 100,000 [1,2,3,4,5]. Rare cancers are defined by the United States National Cancer Institute as those with an incidence fewer than 15 per 100,000 persons per year [6,7], placing pNETs in this category. Patients with pNETs have a mortality rate that has increased roughly fivefold over the past 30–40 years [1]. pNETs are categorized as functional tumors based on clinical symptoms manifested by tumor-secreted hormones [4]. According to the WHO 2019 classification, prognostic factors for gastroenteropancreatic neuroendocrine neoplasms (GEP-NENs) include the mitotic count and Ki-67 index. The resection of primary functional well-differentiated NETs in patients with liver metastases may improve survival [8]. Specifically for pNETs, ~90% are non-functional tumors, which are often asymptomatic and therefore likely to remain undiagnosed until they are advanced and unresectable [9]. As a result, non-functional pNETs have a worse prognosis. At initial diagnosis, 60–70% of pNET patients have liver metastases, which, when resectable, have a high rate of recurrence [10,11,12]. The activation of the mammalian target of the rapamycin (mTOR) pathway in pNETs has been targeted with the mTOR inhibitor everolimus [13,14]. The multiple receptor tyrosine kinase (RTK) inhibitor sunitinib has also been used for pNET treatment [13,14]. Nonetheless, the response to these agents is often limited to stable disease, and resistance invariably occurs with tumor progression [4,12]. Accordingly, the identification of targeted agents for pNET treatment represents an unmet need, which was the motivation for performing the present work.

The *MUC1* gene evolved in mammals to protect barrier tissues from biotic and abiotic insults that result in a loss of homeostasis [15,16,17,18]. *MUC1* encodes a C-terminal (MUC1-C) subunit which is activated by the loss of homeostasis and drives inflammatory, proliferative, and remodeling pathways associated with the wound-healing response [15,16,17,18]. As an adverse consequence of this protective function, the prolonged activation of MUC1-C in settings of chronic inflammation drives cancer progression [17,18]. MUC1-C thus contributes to lineage plasticity and the epithelial–mesenchymal transition (EMT) [16,18]. MUC1-C regulates the (i) Polycomb Repressive Complex 1 (PRC1) and PRC2 [19], (ii) SWI/SNF BAF and PBAF chromatin remodeling complexes [20,21,22], and (iii) COMPASS family of H3K4 methyltransferases [23]. In this way, MUC1-C regulates epigenetic reprogramming and chromatin accessibility across the genomes of cancer cells in driving the cancer stem cell (CSC) state [18]. A dependence on MUC1-C for CSC self-renewal capacity and tumorigenicity has been uncovered across pan-cancers, which are largely adenocarcinomas, as well as those with neuroendocrine (NE) dedifferentiation [16,18,24].

There is no reported involvement of MUC1-C in pNET progression. The present work was performed to determine if patients with these rare cancers are potential candidates for treatment with the anti-MUC1-C agents under development [16,18,24]. We show that *MUC1* expression is upregulated in primary pNETs that progress with metastatic disease. Of functional significance, silencing MUC1-C in pNET cells suppresses the E2F, MYC, and mTOR pathways, which have been associated with proliferative pNETs that have a poor prognosis [25]. In concert with driving the pNET CSC state, we demonstrate that MUC1-C is necessary for NOTCH2 expression, self-renewal capacity, and tumorigenicity. We also report that MUC1-C expression in primary pNET tissues is associated with aggressive disease and a poor prognosis. These findings demonstrate that MUC1-C is of importance in pNET progression and is a potential target for advancing pNET treatment.

## 2. Methods

### 2.1. Analysis of Human pNET Tumor Datasets

Data analysis was performed using the Gene Expression Omnibus (GSE178398) dataset [26].

### 2.2. Cell Culture

QGP-1 cells were maintained in Gibco RPMI 1640 Media containing 10% heat-inactivated FBS and 1% L-glutamine. BON-1 cells were maintained in Gibco Dulbecco’s Modified Eagle Medium Nutrient Mixture F-12 (DMEM/F-12) containing 10% heat-inactivated FBS and 2% L-glutamine. The cells were cultured for 3–4 months. The authentication of the cells was performed by short tandem repeat analysis. The cells were monitored for mycoplasma contamination using the MycoAlert Mycoplasma Detection Kit (Lonza, Rockland, ME, USA).

### 2.3. Gene Silencing and Rescue

MUC1shRNA (MISSION shRNA TRCN0000122938), MUC1shRNA#2 (MISSION shRNA TRCN0000430218), MYCshRNA (MISSION shRNA TRCN0000039642), and a control scrambled shRNA (CshRNA) (Millipore Sigma; Burlington, MA, USA) were inserted into pLKO.1-puro (Plasmid #8453; Addgene, Cambridge, MA, USA) or pLKO-tet-puro (Plasmid #21915; Addgene, Cambridge, MA, USA) as described [27]. Single guide RNAs targeting NOTCH2 were inserted into the lentiCRISPR v2 hygro (Plasmid #98291; Addgene) as described [28]. The viral vectors were produced in 293T cells as described [27]. Flag-tagged MUC1-CD was inserted into pInducer20 (plasmid #44012, Addgene) as described [27]. Cells transduced with the vectors were selected for growth in 1–3 μg/mL puromycin, 100 to 400 mg/mL hygromycin, or 200 to 500 µg/mL geneticin. For tet-inducible vectors, the cells were treated with 0.1% DMSO as the vehicle control or 500 ng/mL doxycycline (DOX; Millipore Sigma).

### 2.4. Real-Time Quantitative Reverse-Transcription PCR (qRT-PCR)

Total RNA was isolated using TRIzol (Invitrogen, Carlsbad, CA, USA). cDNAs were synthesized and amplified as described [27]. The primers used for qRT-PCR are listed in Appendix A.

### 2.5. Immunoblot Analysis

The total lysates prepared from the subconfluent cells were subjected to immunoblot analysis using anti-MUC1-C (HM-1630-P1ABX, 1:1000 dilution; Thermo Fisher Scientific; Waltham, MA, USA), anti-β-actin (A5441, 1:5000 dilution; Sigma-Aldrich; Burlington, MA, USA), anti-MYC (ab32072, 1:1000 dilution; Abcam; Waltham, MA, USA), anti-CCNA2/Cyclin A2 (4656, 1:1000 dilution; Cell Signaling Technology (CST), Danvers, MA, USA), anti-Cyclin B1 (4138, 1:1000 dilution; CST), anti-p-mTOR (5536, 1:1000 dilution; CST), anti-mTOR (2983, 1:1000 dilution; CST), anti-NOTCH2 (5732, 1:1000 dilution; CST), anti-Jagged 1 (2620, 1:1000 dilution; CST), and anti-Histone H3 (ab1791, 1:5000 dilution; Abcam).

### 2.6. Cell Fractionation

A subcellular protein fractionation kit (Thermo Fisher Scientific, 78840) was used to isolate chromatin fractions according to the manufacturer’s instructions.

### 2.7. Coimmunoprecipitation Studies

Coimmunoprecipitation was performed using the Pierce™ Classic Magnetic IP/Co-IP Kit (Thermo Fisher Scientific) with anti-MUC1-C (#MA5-11202; Thermo Fisher Scientific).

### 2.8. RNA-seq Analysis

The total RNA from the cells cultured in triplicate was isolated using the RNeasy Plus Mini Kit (Qiagen; Beverly, MA, USA). TruSeq Stranded mRNA (Illumina, San Diego, CA, USA) was used for library preparation, as described [29]. Raw sequencing reads were aligned to the human genome (GRCh38.74) using STAR. Raw feature counts were normalized and subjected to differential expression analysis using DESeq2. The differential expression rank order was utilized for subsequent GSEA, performed using the fgsea (v1.8.0) package in R. The gene sets queried included those available through the Molecular Signatures Database (MSigDB).

### 2.9. Colony Formation Assays

The cells (1–3 × 10^4^) were seeded in 24-well plates for 24 h and then treated with (i) 0.1% DMSO or 500 ng/mL DOX and (ii) PBS or GO-203. After 7–14 days, the cells were stained with 0.5% crystal violet (LabChem, Zelienople, PA, USA) in 25% methanol. Growth was quantified at 590 nm using a spectrophotometer and normalized to DMSO treatment.

### 2.10. Tumorsphere Formation Assays

The cells (1–3 × 10^4^) were seeded per well in six-well ultra-low attachment culture plates (Corning Life Sciences) in DMEM/F12 50/50 medium (Corning Life Sciences; Tewksbury, MA, USA) with 20 ng/mL EGF (Millipore Sigma), 20 ng/mL bFGF (Millipore Sigma), and a 1% B27 supplement (Gibco), as described [27]. In certain studies, cells were (i) treated with a vehicle or 500 ng/mL DOX and (ii) left untreated or treated with GO-203. Tumorspheres were counted under an inverted microscope in triplicate wells.

### 2.11. Mouse Tumor Model Studies

Six-to-eight-week-old nude mice (The Jackson Laboratory; Bar Harbor, ME, USA) were injected subcutaneously in the flank with 5 × 10^6^ QGP-1 cells in 100 μL of a 1:1 solution of the medium and Matrigel (BD Biosciences; Woburn, MA, USA). When the mean tumor volume reached 150–200 mm^3^, the mice were pair-matched into groups. The mice were treated intraperitoneally each day with PBS or GO-203 at a dose of 12 μg/g body weight. Unblinded tumor measurements and body weights were recorded twice each week. The mice were sacrificed when the tumors reached >2000 mm^3^, as calculated by the following formula: (width)^2^ × length/2. These studies were conducted in accordance with the ethical regulations required for approval by the Dana-Farber Cancer Institute Animal Care and Use Committee under protocol #03-029.

### 2.12. Immunohistochemistry (IHC)

We retrospectively examined tumor tissue samples from patients with pNETs who underwent surgical resection at the Department of Surgery, School of Medicine, Keio University from March 1986 to January 2023. The specimens were subjected to IHC with an anti-MUC1-C rabbit monoclonal antibody (16564, 1:1000 dilution; CST, heat-induced epitope retrieval, pH 6.0). The determinations were performed independently by three investigators (H.O., K.F., and Y.M.). If the independent assessments were not in agreement, the slides were reviewed together by the three investigators until they reached a consensus. The consensus judgments were adopted as the results. Three levels were defined according to the size of the area stained in one specimen (0% = 0; 0–<25% = 1; >25% = 2). The pancreatic ductal epithelium and acinar cells were used as internal positive controls for MUC1-C staining. All pathological materials available for tumor classification were reviewed by expert pathologists using the standard World Health Organization classifications. The disease status was staged according to the TNM staging system (UICC Ver. 8). The ethics committee of Keio University School of Medicine approved this study. Informed consent or a suitable substitute was obtained from the patients in the study. Disease-free survival was measured as the time between the date of operation to the date of recurrence or death from any cause or the date of the last clinical follow-up. Survival rates were calculated using the Kaplan–Meier method, and the difference between curves was assessed using the log-rank test.

### 2.13. Statistical Analysis

Each experiment was performed at least three times. Unpaired two-tailed Student’s *t*-tests were used to assess differences between the mean ± SD of two groups. *p*-values were considered significant at *p* < 0.05. Asterisks represent * *p* ≤ 0.05, ** *p* ≤ 0.01, *** *p* ≤ 0.001, and **** *p* ≤ 0.0001 with CI = 95%.

### 2.14. Data Availability

The accession numbers for the RNA-seq data are GEO Submission GSE267722.

## 3. Results

### 3.1. pNET Cells Are Dependent on MUC1-C for Survival

An analysis of the GSE178398 dataset derived from 22 primary pNET lesions demonstrated that MUC1 mRNA levels are significantly higher in those that progress with metastatic disease (Figure 1A; Appendix A). We therefore analyzed MUC1 expression in the (i) QGP-1 pNET cell line isolated from the primary lesion of a patient with liver metastases [30] and (ii) BON-1 pNET cell line derived from a primary tumor with metastases to lymph nodes [31]. The oncogenic MUC1-C subunit is expressed as an N-glycosylated ~25 kDa glycoprotein and an unglycosylated 17 kDa protein [18,32]. An analysis of QGP-1 and BON-1 cells identified comparable levels of MUC1-C transcripts (Appendix A; Appendix A). An analysis of total cell lysates further demonstrated the predominant expression of the MUC1-C ~25 kDa glycoprotein (Figure 1B). To explore the potential involvement of MUC1-C, we established QGP-1 and BON-1 cells transfected with a control tet-CshRNA or a tet-MUC1shRNA. DOX treatment of QGP-1/tet-MUC1shRNA and BON-1/tet-MUC1shRNA cells downregulated MUC1-C mRNA (Figure 1C) and protein (Figure 1D) levels. By contrast, DOX treatment of QGP-1/tet-CshRNA and BON-1/tet-CshRNA cells had no apparent effect on MUC1-C expression (Appendix A). Of potential translational relevance, we found that silencing MUC1-C in QGP-1 (Figure 1E) and BON-1 (Appendix A) cells suppresses their capacity for clonogenic survival. As a confirmation of MUC1-C dependence, we rescued MUC1-C silencing with the DOX-inducible expression of a tet-Flag-MUC1-C cytoplasmic domain (tet-Flag-MUC1-CD) vector (Figure 1F; Appendix A), which reversed the loss of clonogenicity (Figure 1G; Appendix A). These findings demonstrate that pNET cells are dependent on MUC1-C for survival.

### 3.2. MUC1-C Regulates pNET Cell Transcriptomes

RNA-seq performed on QGP-1 cells demonstrated that MUC1-C silencing results in the downregulation of 2679 genes and the upregulation of 2679 genes (Figure 2A). By comparison, silencing MUC1-C in BON-1 cells was associated with the downregulation and upregulation of 2369 and 2446 genes, respectively (Figure 2A). The GSEA of the QGP-1 and BON-1 datasets (Appendix A) demonstrated that silencing MUC1-C is significantly associated with the suppression of the HALLMARK E2F TARGETS signature (Figure 2B). E2F target genes regulate cancer cell proliferation, genomic integrity, and metabolism [33]. Among 27 common downregulated E2F target genes in QGP-1 and BON-1 cells with MUC1-C silencing (Figure 2C; Appendix A), we identified those encoding chromatin proteins: (i) HMGA1, a driver of the stem cells, inflammatory pathway, and cell cycle genes [34], and (ii) HMGB3, an effector of cell proliferation, self-renewal, and drug resistance (Figure 2D) [35]. Poor prognosis proliferation type pNETs have an enrichment of cell cycle-related gene sets [25]. Consistent with that enrichment, GSEA demonstrated that silencing MUC1-C in QGP-1 and BON-1 cells is significantly associated with the suppression of the BENPORATH CYCLING GENES signature (Figure 2E). Among these genes, we identified 66 that were downregulated in QGP-1 and BON-1 cells with MUC1-C silencing (Figure 2F; Appendix A), which included (i) aurora kinase B (AURKB), a regulator of mitotic cell cycle progression and a potential target for cancer treatment [36], (ii) BUB3, which regulates the mitotic spindle assembly checkpoint [37], and (iii) RAN GTPase, which functions in nucleocytoplasmic transport and cell cycle progression [38] (Figure 2G). These results collectively uncovered a role for MUC1-C in regulating E2F target genes involved in pNET cell cycle progression.

### 3.3. MUC1-C Regulates MYC in pNET Cells

MYC regulates the induction of E2F target genes [39] and is commonly dysregulated in pNET tumors [25]. We found that silencing MUC1-C in QGP-1 and BON-1 cells decreases MYC mRNA (Figure 3A) and protein (Figure 3B) levels. The MUC1-C cytoplasmic domain (MUC1-CD) is an intrinsically disordered protein that integrates diverse signaling pathways (Figure 3C) [18]. MUC1-CD is a substrate for EGFR, FGFR3, and MET phosphorylation [18]. The MUC1-CD CQC motif, which is targeted by the GO-203 inhibitor, binds directly to TCF4 [18]. Additionally, the MUC1-CD SAGNGGSSLS region associates with beta-catenin, which, together with TCF4, induces *MYC* expression (Figure 3C) [18]. The MUC1-C CQC motif also binds directly to the MYC HLH-LZ domain in regulating MYC target genes [40]. We found that the rescue of MUC1-C silencing with MUC1-CD reverses the downregulation of MYC expression (Figure 3D). The GSEA of the QGP-1 and BON-1 gene sets further identified the involvement of MUC1-C in regulating the HALLMARK MYC TARGETS V1 signature (Figure 3E) and common sets of MYC target genes (Figure 3F; Appendix A). To extend these results, we established QGP-1 and BON-1 cells expressing a tet-MYCshRNA, which responded to DOX treatment with the downregulation of MYC, as well as MUC1-C, expression (Figure 3G; Appendix A). In support of a MUC1-C/MYC auto-regulatory pathway, (i) silencing MUC1-C with different MUC1shRNAs to exclude off-target effects (Appendix A) and (ii) silencing MYC (Appendix A) decreased the expression of the cyclin A2 and cyclin B1 proteins that regulate entry into mitosis. Collectively, these results indicate that the QGP-1 and BON-1 cells are dependent on MUC1-C for the activation of E2F and MYC target genes that drive proliferation.

### 3.4. MUC1-C/MYC Signaling Regulates the mTORC1 Pathway

The mammalian target of rapamycin complex 1 (mTORC1) promotes cancer cell growth and survival [41]. The dysregulation of mTOR has been identified in pNET tumors as a target for treatment with everolimus [25,42]. Here, we found that inducible and stable MUC1-C silencing in QGP-1 and BON-1 cells results in the downregulation of p-mTOR (Ser-2448) and mTOR levels (Figure 4A; Appendix A), which were rescued by MUC1-CD expression (Figure 4B). GSEA further demonstrated that silencing MUC1-C in QGP-1 and BON-1 cells is significantly associated with the suppression of the HALLMARK MTORC1 SIGNALING gene signature (Figure 4C). Common mTORC1 signaling genes in QGP-1 and BON-1 cells with MUC1-C silencing included those that regulate (i) glycolysis, such as fructose-1,6-bisphosphate aldolase (ALDOA), enolase 1 (ENO1), and lactate dehydrogenase A (LDHA), and (ii) serine metabolism involving phosphoglycerate dehydrogenase (PHGDH) and phosphoserine aminotransferase (PSAT1) (Figure 4D; Appendix A). mTORC1 is necessary for MYC-driven cancer cell survival [41,43,44,45,46,47]. Along these lines, we found that, like MUC1-C, silencing MYC decreases p-mTOR (Ser-2448) and mTOR expression (Figure 4E). In addition, as found for MUC1-C, silencing MYC suppressed QGP-1 (Figure 4F) and BON-1 (Figure 4G) colony formation, indicating that MUC1-C/MYC signaling regulates effectors of the mTORC1 metabolic pathway in association with driving survival.

### 3.5. MUC1-C/MYC Signaling Integrates the NOTCH Pathway and Self-Renewal Capacity

In searching for other pathways regulated by MUC1-C in QGP-1 and BON-1 cells, we found that MUC1-C silencing is associated with the suppression of the REACTOME SIGNALING BY the NOTCH gene signature (Figure 5A). NOTCH signaling is conferred by the NOTCH1-4 TFs that drive stemness and the CSC state [48]. Silencing MUC1-C in QGP-1 and BON-1 cells decreased the expression of NOTCH2 and the downstream NOTCH pathway effector JAG1 that contributes to stemness (Figure 5B) [48]. Silencing MYC similarly downregulated NOTCH2 (Figure 5C), indicating that the MUC1-C/MYC pathway regulates NOTCH2 signaling. In addition, we confirmed that the downregulation of NOTCH2 expression by MUC1-C silencing is rescued with MUC1-CD (Figure 5D). In concert with the involvement of NOTCH2 signaling in contributing to the CSC state [48], silencing MUC1-C in QGP-1 and BON-1 cells suppressed self-renewal capacity, as determined by tumorsphere formation (Figure 5E; Appendix A). As confirmation of MUC1-C dependence, rescuing MUC1-C silencing with MUC1-CD restored the capacity for self-renewal (Figure 5E; Appendix A). Silencing MYC in QGP-1 and BON-1 cells also suppressed self-renewal capacity (Figure 5F; Appendix A). In extending these results, we found that targeting NOTCH2 decreases tumorsphere formation (Figure 5G; Appendix A), confirming that MUC1-C/MYC signaling regulates NOTCH2 and, with it, self-renewal capacity.

### 3.6. Targeting MUC1-C with the GO-203 Inhibitor Suppresses MUC1-C/MYC Signaling, Self-Renewal, and Tumorigenicity

MUC1-CD is an intrinsically disordered 72 aa protein that includes a CQC motif necessary for the formation of MUC1-C homodimers and their import into the nucleus [16]. The cell-penetrating GO-203 peptide (D-amino acids: R_9_-CQCRRKN) targets the CQC motif with the selective dose-dependent inhibition of MUC1-C function in vitro and in vivo [16]. As found in other types of cancer cells, MUC1-C is expressed as 17 kDa and higher-order structures in chromatin from QGP-1 and BON-1 cells (Appendix A), which was decreased by GO-203 treatment (Appendix A) [49,50]. Consistent with the involvement of the MUC1-C CQC motif in direct binding to MYC [40], GO-203 abrogated the formation of MUC1-C/MYC heterodimers and the localization of MYC in chromatin (Appendix A–D). In this way, GO-203 (i) decreased the expression of MUC1-C/MYC target genes encoding cyclin A2, cyclin B1, mTOR, and NOTCH2 (Figure 6A), (ii) inhibited colony formation (Figure 6B,C), and (iii) suppressed tumorsphere formation (Figure 6D,E). Furthermore, GO-203 treatment of established QGP-1 xenografts in nude mice inhibited tumorigenicity in association with the suppression of MYC expression (Figure 6F,G). These results confirmed that QGP-1 and BON-1 cells are dependent on MUC1-C for the regulation of MYC and downstream effectors, which drive proliferation, survival, and self-renewal capacity.

### 3.7. Association of MUC1-C Expression in pNET Tumors with Adverse Clinical Outcomes

Based on the findings that pNET cell lines are MUC1-C-dependent, we analyzed the expression of MUC1-C by IHC in a cohort of surgically resected primary pNETs from 58 patients (Figure 7A; Appendix A). Consistent with the analysis of MUC1 mRNA levels in the GSE178398 dataset (Figure 1A), the incidence of MUC1-C positivity was significantly higher in primary tumors that progress with metastases than in those limited to localized disease (63% vs. 10%, *p* = 0.002) (Figure 7B; Table 1). MUC1-C expression was also significantly associated with the NET grade (G2-3, *p* = 0.004) and advanced pathological stage (II-IV, *p* = 0.001) (Table 1). Survival analyses by the Kaplan–Meier method further showed that patients with MUC1-C-positive tumors have a significantly shorter disease-free survival (*p* = 0.0022) (Figure 7C), indicating that primary pNETs expressing MUC1-C exhibit aggressive characteristics and a poor prognosis.

## 4. Discussion

Rare cancers are defined in the United States and Europe as those with fewer than 6–15 cases per 100,000 people per year [7]. pNETs fall within this definition of a rare cancer, with an incidence of approximately 1 per 100,000 per year [5]. The identification of druggable targets for pNET treatment has been largely limited to mTOR and RTKs [13,14]. Unfortunately, targeting mTOR with everolimus and RTKs with sunitinib has had limited effectiveness in the treatment of metastatic pNETs [13,14], emphasizing a need for identifying other potential targets. Funding for the research and treatment of pNETs and other rare cancers is woefully limited as compared to that for common malignancies [51]. Moreover, clinical trials for patients with these rare cancers are often challenged by slower rates of accrual and a lack of access to targeted agents that might be appropriate for their treatment [51].

In considering these challenges for patients with advanced pNETs, the present work focused on the MUC1-C oncoprotein, which has been uncovered as a pan-cancer druggable target [18]. Research on MUC1-C initially focused on adenocarcinomas, and those findings were extended to cancers with NE dedifferentiation, including neuroendocrine prostate cancer (NEPC), small cell lung cancer (SCLC), and Merkel Cell Cancer (MCC) [24,28,29,52,53]. We found that MUC1 expression is upregulated in primary pNETs that progress with metastatic disease. Studies of the QGP-1 and BON-1 cell lines further demonstrated that MUC1-C is necessary for survival. Our previous work in NEPC, SCLC, and MCC cell models revealed that their addiction to MUC1-C is associated with the dysregulation of the E2F and MYC pathways [24,28,29,52,53]. By extension to pNET cells, we found that MUC1-C also regulates E2F and MYC signaling pathways (Figure 7D), which contribute to uncontrolled proliferation in cancer cells [33,39]. These findings, like those uncovered in other NE cancers [24,28,29,52,53], indicated that pNET cells are dependent on MUC1-C-driven oncogenic functions.

The dysregulation of MYC and mTORC1 has been widely identified in pNET tumors by mechanisms that have remained unclear [25]. Our studies demonstrate that MUC1-C/MYC signaling is necessary for mTOR expression in pNET cells and that targeting MUC1-C suppresses mTOR activation. The findings that pNET cells are dependent on MUC1-C for the activation of MYC and mTOR thus further supported their addiction to MUC1-C for clonogenic survival. MUC1-C localizes to chromatin [49], where it interacts with MYC and effectors of epigenetic reprogramming [19,20,21,22,23]. MUC1-C thereby regulates epigenetic reprogramming and chromatin accessibility in driving the CSC state [18]. Remarkably little is known about pNET CSCs and the underlying pathways that contribute to their self-renewal. Along these lines, our results demonstrate that MUC1-C is necessary for pNET cell self-renewal and the CSC state (Figure 7D). Similar findings have been reported in other NE cancers [24,28,29,52,53], whereas the dependence on MUC1-C signaling identified here has uncovered a pathway responsible for the dysregulation of MYC and mTOR in pNET cells.

In further support of MUC1-C as a novel target for pNET treatment, we found that *MUC1* gene expression is upregulated in primary pNET tumors that progress with metastatic disease. We also found that MUC1-C (i) is upregulated in primary pNETs from patients with metastatic disease and (ii) is associated with more aggressive characteristics, including the NET grade and pathological grade, and poor clinical outcomes. Of translational relevance, targeting MUC1-C with the GO-203 inhibitor suppressed pNET cell survival, self-renewal capacity, and tumorigenicity (Figure 7D). Antisense oligonucleotides (ASOs) have also been generated to target the MUC1-C cytoplasmic domain [54]. In addition, an antibody generated against the MUC1-C extracellular domain has been advanced for clinical evaluation as CAR-T cells (Poseida Pharmaceuticals) and is being developed as an antibody–drug conjugate by the NCI NExT Program for MUC1-C-expressing cancers [55].

In summary, our results demonstrate that pNETs are addicted to MUC1-C by the activation of pathways that promote the CSC state. These findings are of potential therapeutic importance in having identified MUC1-C as a novel target for pNET treatment with the anti-MUC1-C agents that are under clinical development.

## Figures and Tables

**Figure 1 biomedicines-12-01509-f001:**
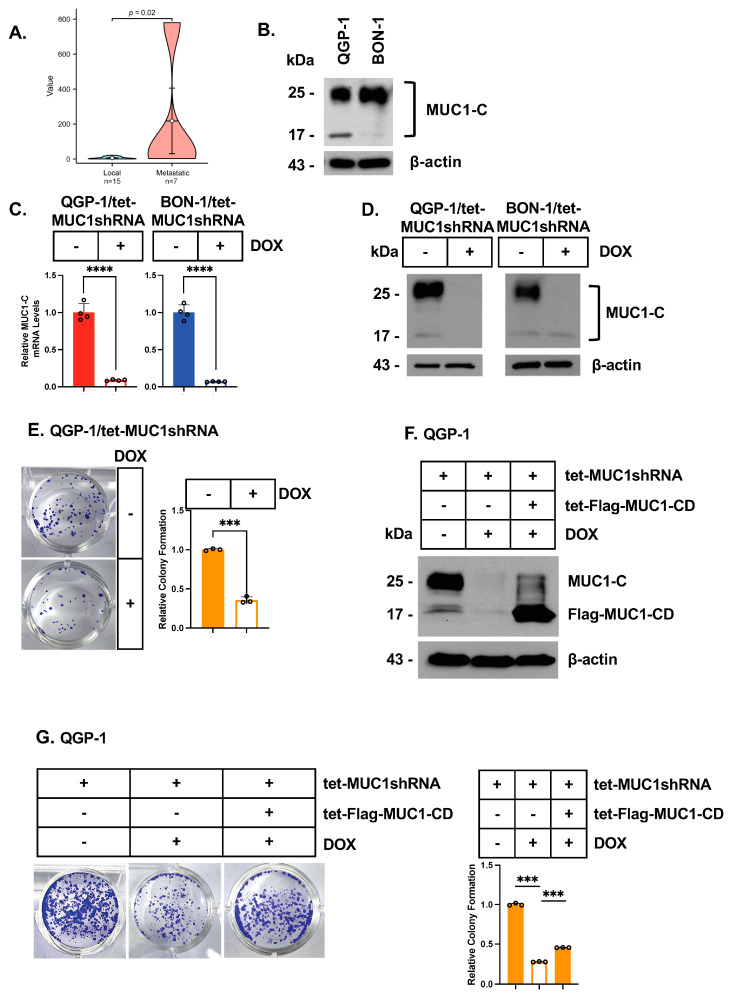
**QGP-1 and BON-1 cells are dependent on MUC1-C for clonogenic survival.** (**A**) Analysis of the GSE178398 pNET dataset demonstrating that MUC1 expression is significantly upregulated in primary tumors from patients with metastatic vs. localized disease. (**B**) Total cell lysates from QGP-1 and BON-1 cells were immunoblotted with antibodies against the indicated proteins. (**C**,**D**) QGP-1/tet-MUC1shRNA and BON-1/tet-MUC1shRNA cells treated with the vehicle or DOX for 7 days were analyzed for MUC1-C transcripts by qRT-PCR using the primers listed in Appendix A. The results (mean ± SD of four determinations) represent relative levels compared to vehicle-treated cells (assigned a value of 1) (**C**). Lysates were analyzed by immunoblotting with antibodies against the indicated proteins (**D**). (**E**) QGP-1/tet-MUC1shRNA cells treated with the vehicle or DOX for 7 days were analyzed for colony formation. Photomicrographs of representative stained colonies (left) are shown. The results (mean ± SD of three determinations) are expressed as colony formation relative to that for vehicle-treated cells (assigned a value of 1) (right). (**F**) QGP-1 cells expressing the indicated vectors were treated with the vehicle or DOX for 7 days and then analyzed by immunoblotting with antibodies against the indicated proteins. (**G**) QGP-1 cells expressing the indicated vectors were treated with the vehicle or DOX for 7 days and analyzed for colony formation. Photomicrographs of representative stained colonies are shown. The results (mean ± SD of three determinations) are expressed as colony formation relative to that for vehicle-treated cells (assigned a value of 1).

**Figure 2 biomedicines-12-01509-f002:**
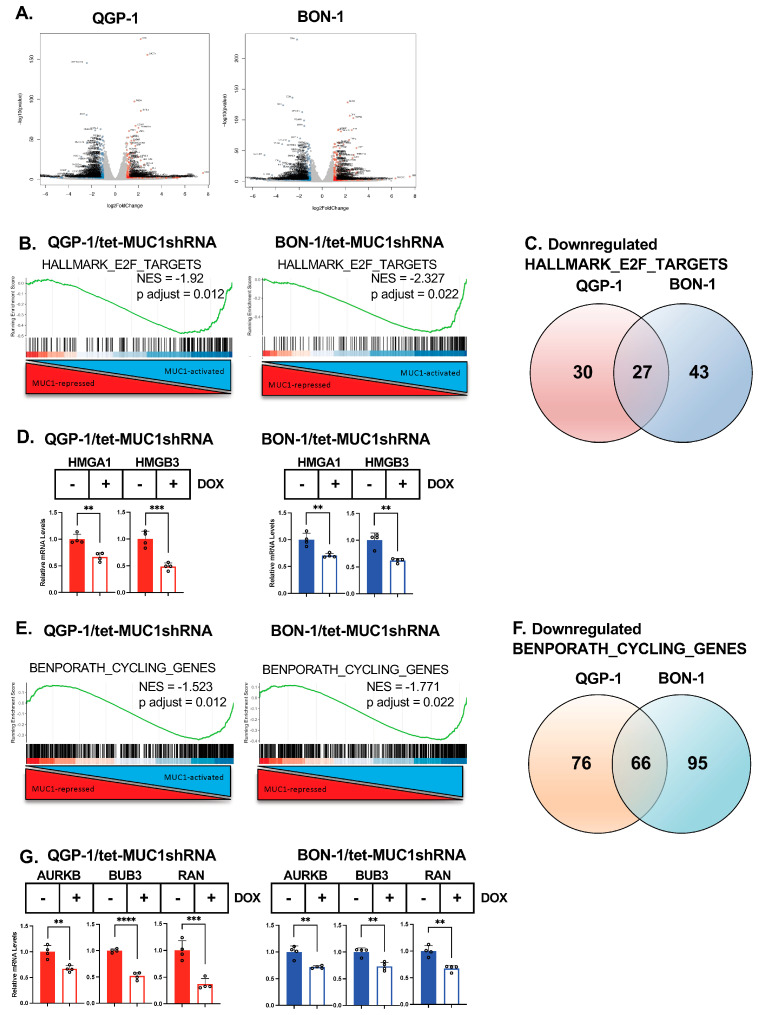
**Silencing MUC1-C in pNET cells downregulates E2F and MYC target gene and cell cycle gene signatures.** (**A**) RNA-seq was performed on biologic triplicates of QGP-1/tet-MUC1shRNA and BON-1/tet-MUC1shRNA cells treated with the vehicle or DOX for 7 days. Volcano plots depicting downregulated (left) and upregulated (right) genes with MUC1-C silencing. (**B**) GSEA of the QGP-1 and BON-1 RNA-seq datasets using the HALLMARK E2F TARGET gene signature. (**C**) Venn diagram of downregulated HALLMARK E2F TARGET genes in QGP-1 and BON-1 cells with MUC1-C silencing. (**D**) QGP-1 and BON-1 cells were analyzed for the indicated transcripts by qRT-PCR using the primers listed in Appendix A. The results (mean ± SD of four determinations) are expressed as relative levels compared to those obtained for vehicle-treated cells (assigned a value of 1). (**E**) GSEA of the QGP-1 and BON-1 RNA-seq datasets using the BENPORATH CYCLING GENES signature. (**F**) Venn diagram of downregulated BENPORATH CYCLING GENES in QGP-1 and BON-1 cells with MUC1-C silencing. (**G**) QGP-1 and BON-1 cells were analyzed for the indicated transcripts by qRT-PCR using the primers listed in Appendix A. The results (mean ± SD of four determinations) are expressed as relative levels compared to those obtained for vehicle-treated cells (assigned a value of 1).

**Figure 3 biomedicines-12-01509-f003:**
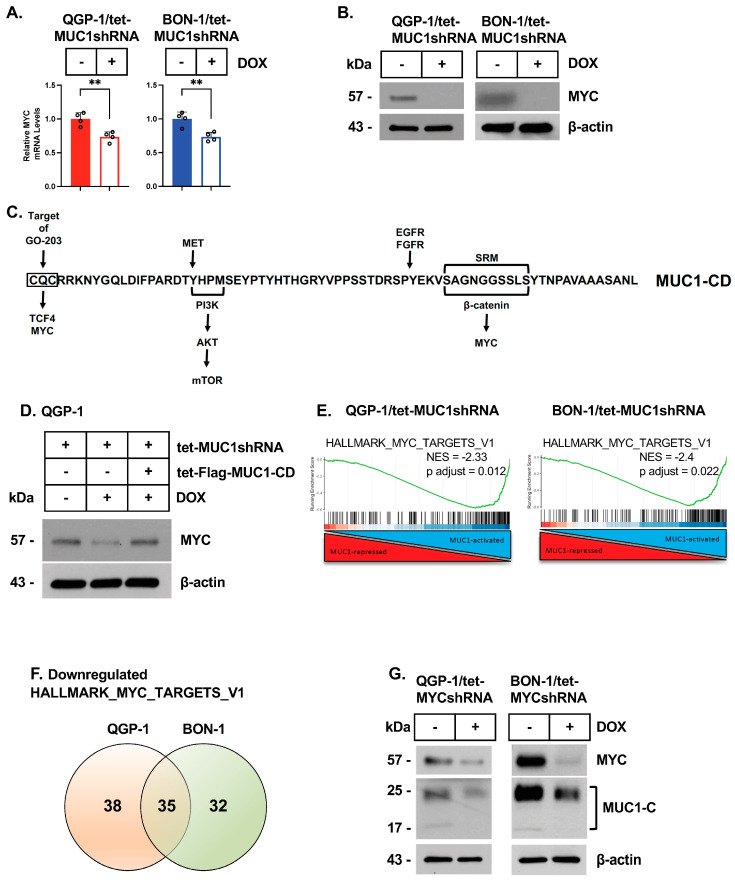
**MUC1-C regulates MYC expression in pNET cells.** (**A**,**B**) QGP-1/tet-MUC1shRNA and BON-1/tet-MUC1shRNA cells were treated with the vehicle or DOX for 7 days and analyzed for MYC transcripts by qRT-PCR (**A**). The results (mean ± SD of four determinations) are expressed as levels relative to those for vehicle-treated cells (assigned a value of 1). Lysates were analyzed by immunoblotting with antibodies against the indicated proteins (**B**). (**C**) Amino acid sequence of the MUC1-C cytoplasmic domain highlighting (i) the direct binding of the CQC motif with MYC and TCF4 and (ii) the interaction of the serine-rich motif (SRM) with beta-catenin. In this way, the MUC1-C cytoplastic domain facilitates the formation of TCF4/beta-catenin complexes in activating the *CCND1* and *MYC* genes. The MUC1-C cytoplasmic domain also includes a YHPM sequence that, when phosphorylated on tyrosine, conforms to a consensus sequence for the binding of the PI3K SH2 domain. (**D**) Lysates from QGP-1 cells expressing the indicated vectors treated with the vehicle or DOX for 7 days were immunoblotted with antibodies against the indicated proteins. (**E**) GSEA of the QGP-1 and BON-1 RNA-seq datasets using the HALLMARK MYC TARGETS V1 gene signature. (**F**) Venn diagram of downregulated HALLMARK MYC TARGETS V1 genes in QGP-1 and BON-1 cells with MUC1-C silencing. (**G**) Lysates from QGP-1/tet-MYCshRNA and BON-1/tet-MYCshRNA cells treated with the vehicle or DOX for 7 days were immunoblotted with antibodies against the indicated proteins.

**Figure 4 biomedicines-12-01509-f004:**
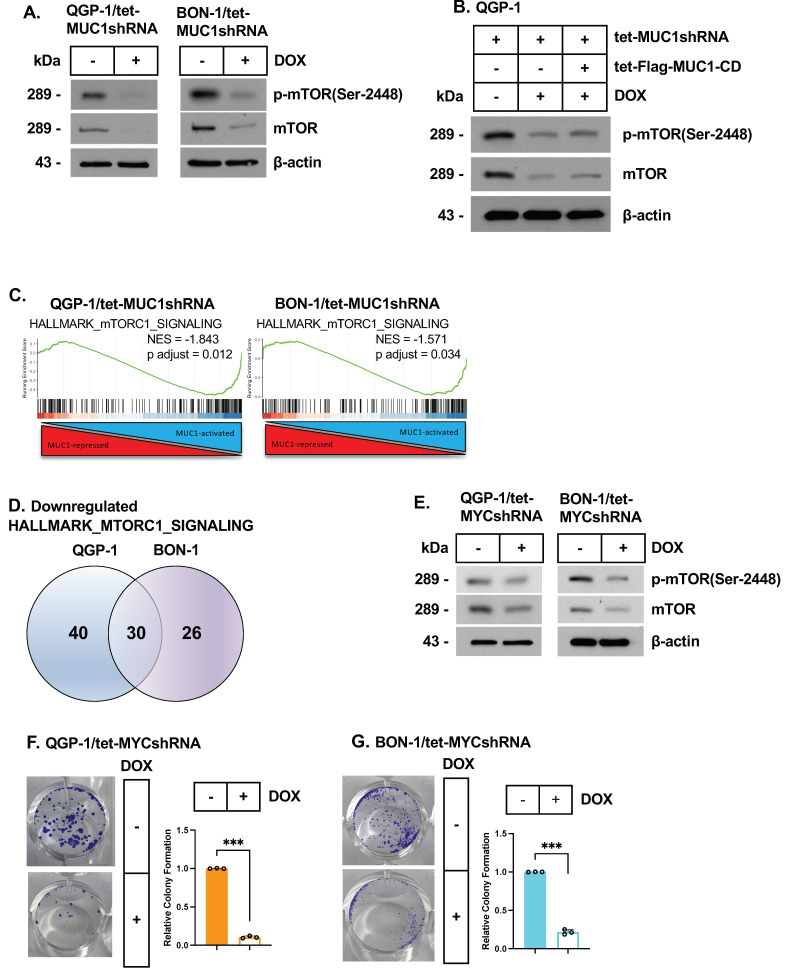
**MUC1-C/MYC signaling regulates the mTOR pathway.** (**A**) QGP-1/tet-MUC1shRNA and BON-1/tet-MUC1shRNA cells were treated with the vehicle or DOX for 7 days and analyzed by immunoblotting with antibodies against the indicated proteins. (**B**) QGP-1 cells expressing the indicated vectors were treated with the vehicle or DOX for 7 days and analyzed by immunoblotting with antibodies against the indicated proteins. (**C**) GSEA of the QGP-1 and BON-1 RNA-seq datasets using the HALLMARK mTORC1 SIGNALING gene signature. (**D**) Venn diagram of downregulated HALLMARK mTORC1 SIGNALING genes in QGP-1 and BON-1 cells with MUC1-C silencing. (**E**) Lysates from QGP-1/tet-MYCshRNA and BON-1/tet-MYCshRNA cells treated with the vehicle or DOX for 7 days were immunoblotted with antibodies against the indicated proteins. (**F**,**G**) QGP-1/tet-MYCshRNA (**F**) and BON-1/tet-MYCshRNA (**G**) cells treated with the vehicle or DOX for 7 days were analyzed for colony formation. Photomicrographs of representative stained colonies (left) are shown. The results (mean ± SD of three determinations) are expressed as the colony formation relative to that for vehicle-treated cells (assigned a value of 1) (right).

**Figure 5 biomedicines-12-01509-f005:**
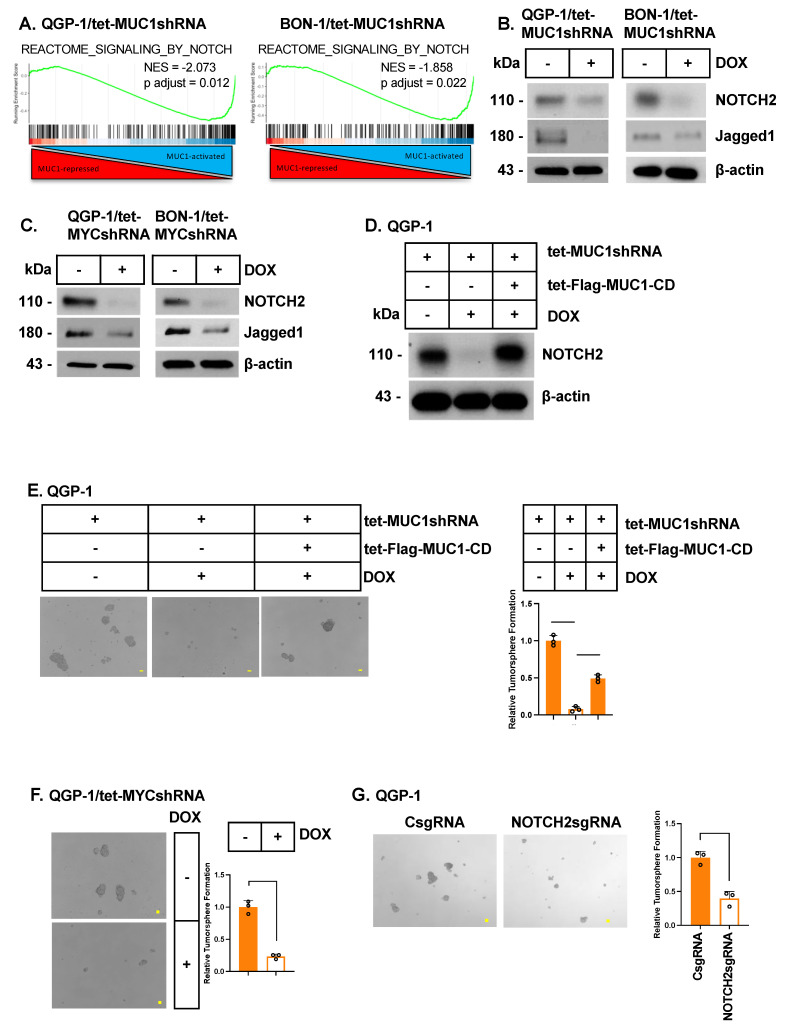
**MUC1-C/MYC signaling integrates the NOTCH2 pathway and self-renewal capacity.** (**A**) GSEA of the QGP-1 and BON-1 RNA-seq datasets using the REACTOME SIGNALING BY NOTCH gene signature. (**B**) QGP-1/tet-MUC1shRNA and BON-1/tet-MUC1shRNA cells were treated with the vehicle or DOX for 7 days and analyzed by immunoblotting with antibodies against the indicated proteins. (**C**) QGP-1/tet-MYCshRNA and BON-1/tet-MYCshRNA cells were treated with the vehicle or DOX for 7 days and analyzed by immunoblotting with antibodies against the indicated proteins. (**D**) Lysates from QGP-1 cells expressing the indicated vectors treated with the vehicle or DOX for 7 days were immunoblotted with antibodies against the indicated proteins. (**E**) Representative images of tumorspheres derived from the indicated QGP-1 cells treated with the vehicle or DOX for 7 days. The bar represents 100 microns. The results (mean ± SD of three determinations) are expressed as the relative sphere formation compared to that for vehicle-treated cells (assigned a value of 1). (**F**) Representative images of tumorspheres derived from the QGP-1/tet-MYCshRNA cells treated with the vehicle or DOX for 7 days. The bar represents 100 microns. The results (mean ± SD of three determinations) are expressed as the relative sphere formation compared to that for vehicle-treated cells (assigned a value of 1). (**G**) Representative images of tumorspheres derived from QGP-1/CsgRNA and QGP-1/NOTCH2sgRNA cells. The bar represents 100 microns. The results (mean ± SD of three determinations) are expressed as the relative sphere formation compared to that for vehicle-treated cells (assigned a value of 1).

**Figure 6 biomedicines-12-01509-f006:**
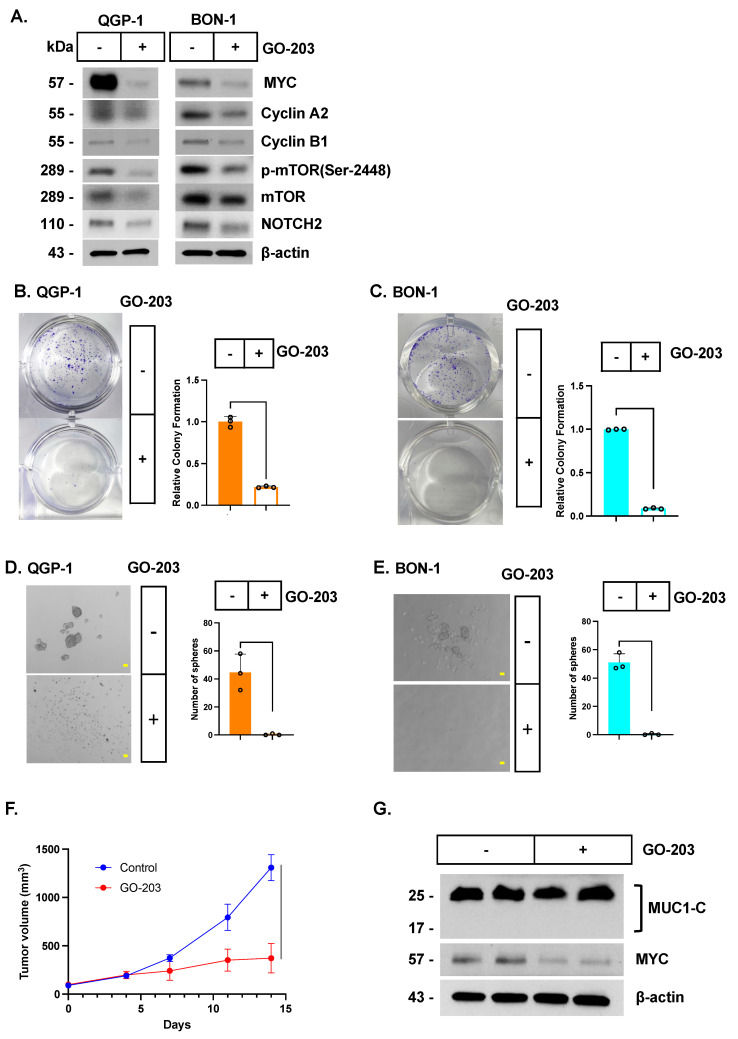
**Targeting MUC1-C with the GO-203 inhibitor suppresses the MUC1-C/MYC pathway, self-renewal, and tumorigenicity.** (**A**) Lysates from QGP-1 cells and BON-1 treated with 3 μM GO-203 for 4 days were immunoblotted with antibodies against the indicated proteins. (**B**,**C**) QGP-1 (**B**) and BON-1 (**C**) cells treated with the vehicle or 5 μM GO-203 for 7 days were analyzed for colony formation. Photomicrographs of representative stained colonies are shown. The results (mean ± SD of three determinations) are expressed as colony formation relative to that for vehicle-treated cells (assigned a value of 1). (**D**,**E**) Representative images of tumorspheres derived from QGP-1 (**D**) and BON-1 (**E**) cells treated with the vehicle or 5 μM GO-203 for 7 days. The bar represents 100 microns. The number of tumorspheres is expressed as the mean ± SD of three determinations. (**F**,**G**) Six-week-old nude mice were injected subcutaneously in the flank with 5 × 10^6^ QGP-1 cells. Mice were pair-matched into two groups when tumors reached 150–200 mm^3^ and treated intraperitoneally each day with PBS or GO-203 at a dose of 12 µg/g body weight. Tumor volumes are expressed as the mean ± SEM of six mice (**F**). Lysates from tumors harvested on day 27 were analyzed by immunoblotting with antibodies against the indicated proteins (**G**).

**Figure 7 biomedicines-12-01509-f007:**
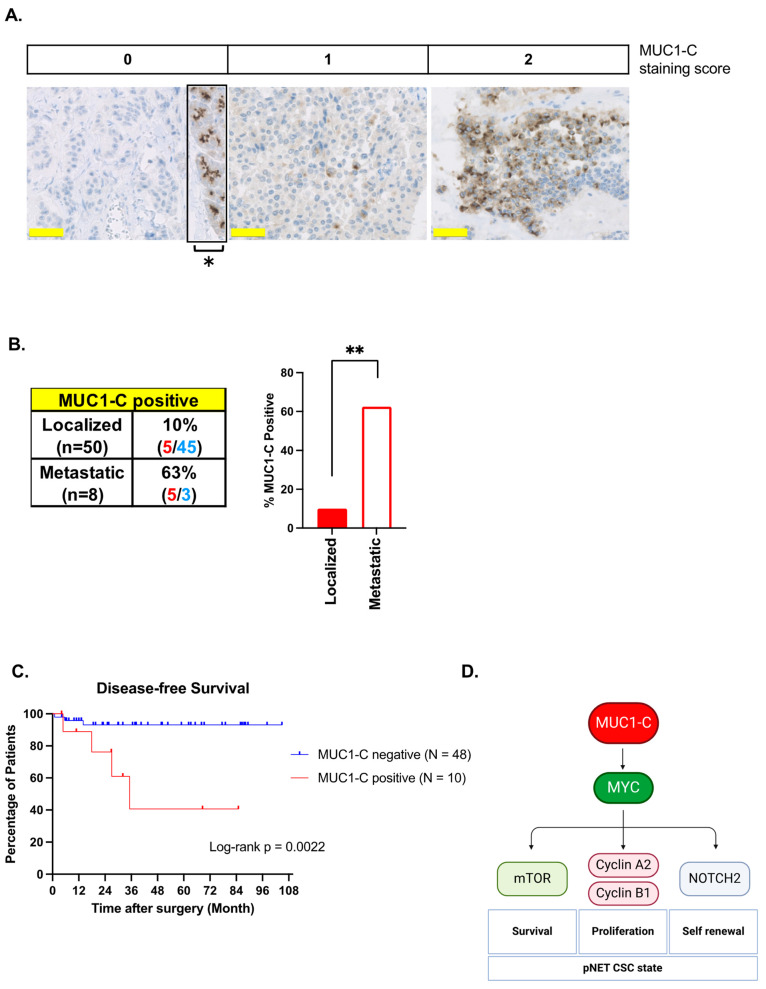
**Association of MUC1-C expression in pNET tumors with adverse clinical outcomes.** (**A**) Immunostaining of MUC1-C in patients with pNET. Representative immunostaining for MUC1-C scoring of 0, 1, and 2 in surgical specimens from patients with pNET are shown as defined according to the size of the stained area (0% = 0; 0–<25% = 1; >25% = 2). As an internal positive control, MUC1-C expression is shown in the insert for non-tumoral acinar cells (asterisks). The bar represents 50 microns. (**B**) Percent of MUC1-C-positive primary pNET tissues from patients with localized and metastatic disease. The asterisk (**) denotes a *p*-value ≤ 0.01. (**C**) Kaplan–Meier curves of disease-free survival in patients with MUC1-C-negative and MUC1-C-positive primary pNET tissues. (**D**) Proposed pNET model based on the findings that MUC1-C (i) drives MYC expression in an auto-inductive pathway and (ii) forms complexes with MYC that regulate MYC target genes. MUC1-C/MYC signaling regulates E2F target genes, such as HMGA1 and HMGB3, that promote self-renewal and proliferation and effectors of cell cycle progression, including AURKB, BUB3, RAN, and cyclins A2/B1. MUC1-C/MYC signaling integrates the regulation of proliferation with the induction of (i) mTOR and survival and (ii) NOTCH2 and self-renewal capacity, which collectively contribute to the pNET CSC state.

**Table 1 biomedicines-12-01509-t001:** Clinical characteristics of patients with pNET tumors stained for MUC1-C expression by IHC.

Factors		Total n = 58	MUC1-C Negative n = 48	MUC1-C Positive n = 10	*p*-Value
Age, years	<70	38	32	6	0.724
	≥70	20	16	4	
Sex	Male	34	28	6	1.000
	Female	24	20	4	
Tumor size (mm)	<20	41	37	4	0.050
	>20	17	11	6	
Grade	G1	31	30	1	0.004
	G2/G3	27	18	9	
Stage (UICC8th)	I	44	41	3	0.001
	II, III, IV	14	7	7	
Metastasis		8	3	5	0.002

## Data Availability

The original contributions presented in the study are included in the article/Appendix A, further inquiries can be directed to the corresponding author.

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
