# Peer review of "MUC1-C Dependence for the Progression of Pancreatic Neuroendocrine Tumors Identifies a Druggable Target for the Treatment of This Rare Cancer"

_biomedicines, 2024, doi:10.3390/biomedicines12071509_

Round 1
Reviewer 1 Report
Comments and Suggestions for Authors
The manuscript is very well written and relevant. My main concern is that it is too long and maybe it should be shortened.
The authors should emphasize in the comments the potential clinical implications of their findings, particularly in light of the current literature.
I would remove the term "rare cancer" from the title because pNETs are not that rare!!!
Since we are talking of advanced disease, the authors should comment the other prognostic factors in these patients, for example primary tumor resection in metastatic disease (in this regard, cite the recent paper: Citterio D, et al Primary tumour resection may improve survival in functional well-differentiated neuroendocrine tumours metastatic to the liver. Eur J Surg Oncol. 2017 Feb;43(2):380-387. )
Author Response
The manuscript is very well written and relevant.
Concern 1: My main concern is that it is too long and maybe it should be shortened.
Response 1: We have attempted where possible to shorten the Introduction, Results and Discussion sections.
Concern 2: The authors should emphasize in the comments the potential clinical implications of their findings, particularly in light of the current literature.
Response 2: In response, we emphasize that patients with metastatic pNETs have limited treatment options and the identification of new targeted agents represents an unmet need according to the following:
Introduction. Response to available agents is often limited to stable disease and resistance invariably occurs with tumor progression. Accordingly, the identification of targeted agents for pNET treatment represents an unmet need, which was the motivation for performing the present work.
Concern 3: I would remove the term "rare cancer" from the title because pNETs are not that rare!
Response 3: We have again reviewed the definition of “rare cancers” in the United States and Europe and the incidence of pNETs and have added the following information:
Introduction. Rare cancers are defined by the United States National Cancer Institute as those with an incidence fewer than 15 per 100,000 persons per year [US NIH National Cancer Institute: Dictionary of Cancer Terms; Rare Cancer, 2021].
Discussion. Rare cancers are defined in the United States and Europe as those with fewer than 6-15 cases per 100,000 people per year [Sandler, 2021, #12894].
pNETs fall within this definition of a rare cancer with an incidence of approximately 1 per 100,000 per year [Dasari, 2017, #12706; Xu, 2021, #12708; Das, 2021, #12707; Maharjan, 2021, #12709; Hopper, 2019, #12872]; accordingly, we have used the term “rare” cancer in the title and text.
Concern 4. Since we are talking of advanced disease, the authors should comment the other prognostic factors in these patients, for example primary tumor resection in metastatic disease (in this regard, cite the recent paper: Citterio D, et al Primary tumour resection may improve survival in functional well-differentiated neuroendocrine tumours metastatic to the liver. Eur J Surg Oncol.2017 Feb;43(2):380-387.
Response 4. In response, we have cited this information and the reference in the Introduction.
Reviewer 2 Report
Comments and Suggestions for Authors
The manuscript by Ozawa et al on the role of MUC1-C in pancreatic neuroendocrine tumors was so elegantly and concisely written, it was a pleasure to read it. Keep up with the good work!
From the starting hypothesis to the experimental design, everything is clean and convincing, and I recommend publication in the present form.
Author Response
Reviewer #2:
No required responses.
Round 2
Reviewer 1 Report
Comments and Suggestions for Authors
The authors improved their manuscript but still some of my previous concerns remained not addressed. Foe xample, the authors should still shorten a little the paper and concern 4 was not addressed at all. I report it again:
Concern 4. Since we are talking of advanced disease, the authors should comment the other prognostic factors in these patients, for example primary tumor resection in metastatic disease (in this regard, cite the recent paper: Citterio D, et al Primary tumour resection may improve survival in functional well-differentiated neuroendocrine tumours metastatic to the liver. Eur J Surg Oncol.2017 Feb;43(2):380-387.
Author Response
Reviewer #1:
The authors improved their manuscript but still some of my previous concerns remained not addressed. For example, the authors should still shorten a little the paper and concern 4 was not addressed at all. I report it again:
Concern 4. Since we are talking of advanced disease, the authors should comment the other prognostic factors in these patients, for example primary tumor resection in metastatic disease (in this regard, cite the recent paper: Citterio D, et al Primary tumour resection may improve survival in functional well-differentiated neuroendocrine tumours metastatic to the liver. Eur J Surg Oncol. 2017 Feb;43(2):380-387.
Response 4. In point of fact, we already cited this reference in the Introduction in the previous revision. We are again highlighting this information and the following response for the reviewer: “According to the WHO 2019 classification, prognostic factors for gastroenteropancreatic neuroendocrine neoplasms (GEP-NENs) include mitotic count and Ki-67 index”.
Regarding length, the total number of words after the first revision is now 3044. (Title page 114, Abstract 148, Introduction-Results 2130, Discussion 652).
For publications in Biomedicines, the only requirement is that the Abstract should be fewer than 200 words.
We respectfully contend that the manuscript was shortened as much as possible in the first revision so as not to detract from the importance of the work.

Round 3
Reviewer 1 Report
Comments and Suggestions for Authors
The revised manuscript is OK. Thank you!
Author Response
No required responses.